# Restriction Endonuclease-Based Assays for DNA Detection and Isothermal Exponential Signal Amplification

**DOI:** 10.3390/s20143873

**Published:** 2020-07-11

**Authors:** Maria Smith, Kenneth Smith, Alan Olstein, Andrew Oleinikov, Andrey Ghindilis

**Affiliations:** 1TORCATECH, LLC, 5210 104th Street SW, Mukilteo, WA 98275, USA; mariyasmit@hotmail.com (M.S.); kencolfax@hotmail.com (K.S.); 2Paradigm Diagnostics, Inc., 800 Transfer Rd #12, St Paul, MN 55114, USA; olstein@comcast.net; 3Department of Biomedical Science, Charles E. Schmidt College of Medicine, Florida Atlantic University, 777 Glades Road, Boca Raton, FL 33428, USA; aoleinikov@health.fau.edu

**Keywords:** DNA assay, nucleic acid, isothermal, signal amplification, restriction endonuclease

## Abstract

Application of restriction endonuclease (REase) enzymes for specific detection of nucleic acids provides for high assay specificity, convenience and low cost. A direct restriction assay format is based on the specific enzymatic cleavage of a target–probe hybrid that is accompanied with the release of a molecular marker into the solution, enabling target quantification. This format has the detection limit in nanomolar range. The assay sensitivity is improved drastically to the attomolar level by implementation of exponential signal amplification that is based on a cascade of self-perpetuating restriction endonuclease reactions. The cascade is started by action of an amplification “trigger”. The trigger is immobilized through a target-specific probe. Upon the target probe hybridization followed with specific cleavage, the trigger is released into the reaction solution. The solution is then added to the assay amplification stage, and the free trigger induces cleavage of amplification probes, thus starting the self-perpetuating cascade of REase-catalyzed events. Continuous cleavage of new amplification probes leads to the exponential release of new triggers and rapid exponential signal amplification. The proposed formats exemplify a valid isothermal alternative to qPCR with similar sensitivity achieved at a fraction of the associated costs, time and labor. Advantages and challenges of the approach are discussed.

## 1. Introduction

Nucleic acid assays of different formats provide a core for modern-day biotechnology and diagnostics. The critically important parameter is the assay specificity since nucleic acid target detection is usually performed in complex samples that contain DNA from different organisms. The detection specificity for most nucleic acid-based assays (qPCR, LAMP, microarrays, etc.) relies on biorecognition events of DNA strand hybridization and can be adversely affected by non-specific DNA–DNA binding. Addition of a second biorecognition event based on Class II restriction endonucleases (REases) has numerous advantages, first and foremost due to the nearly absolute specificity of these enzymes for particular double-stranded (ds) DNA recognition sites. Therefore, for the enzymatic action to take place, a hybridization event has to form a corresponding specific restriction site (usually palindromic with the total length of 4–8 bp) within the DNA double helix [1,2]. Thus, two biorecognition events are involved in signal generation, making it double-proof in terms of specificity and insensitivity to non-specific binding.

The technical principle of REase-based assays is associated with the release of an enzymatic reaction product from solid support into the liquid phase as the result of target–probe complex cleavage. The product quantification can then be done in various ways by transferring the product-containing liquid phase into a separate reaction setup.

In addition to target–probe hybrid recognition, REases can also be used for exponential signal amplification if the initial hybrid cleavage event releases a “trigger” molecule. The trigger molecule is initially attached to the surface through an assay probe, where upon cleavage it is released into the reaction solution. The free trigger can migrate or be transferred to another surface that is modified with special “amplification” DNA probes. Specific trigger interaction with an amplification probe results in enzymatic cleavage of the probe. Each amplification probe carries additional (one or multiple) trigger molecules, thus the probe cleavage provides for the release of new triggers. This self-perpetuating cascade of cleavage events progresses exponentially until the reaction is stopped (or amplification probes are exhausted).

Several types of triggers can be used. Thus far, we have developed the following two approaches. The first is based on trigger REase enzymes that are immobilized through coupling to oligonucleotides, and can specifically cleave double-stranded oligonucleotide amplification probes. The second employs trigger oligonucleotides that hybridize to single-stranded oligonucleotide amplification probes, thus creating double-stranded REase restriction sites and subjecting them to cleavage by corresponding REase present in the solution. Both approaches are discussed in detail below, and they provide for the development of simple, low-cost, isothermal DNA hybridization assay platforms with exponential signal amplification that can achieve sensitivity similar to PCR applications.

The isothermal nucleic acid assay format is critical for the development of point of care units and field assays. One of the first isothermal assays called nucleic acid sequence-based amplification (NASBA) was introduced in 1991 by J. Compton [3]. Numerous other isothermal nucleic acid assays are reviewed in [4] including strand displacement amplification (SDA), loop-mediated amplification (LAMP), invader assay, rolling circle amplification (RCA), signal-mediated amplification of RNA technology (SMART), helicase-dependent amplification (HDA), recombinase polymerase amplification (RPA), nicking endonuclease signal amplification (NESA) and nicking endonuclease-assisted nanoparticle activation (NENNA), exonuclease-aided target recycling, junction or Y-probes, split DNAZyme and deoxyribozyme amplification strategies, template-directed chemical reactions that lead to amplified signals, non-covalent DNA catalytic reactions, hybridization chain reactions (HCR) and detection via the self-assembly of DNA probes to give supramolecular structures. However, all of them have limitations, and none are yet ready to replace PCR-based methods for the current DNA assay market. Our REase-based isothermal DNA assays are novel and promising, and the corresponding advantages and limitations are discussed in the current work. We are presenting our perspective on these novel assay formats and their potential applications.

## 2. Direct Restriction Assay (DRA)

We introduced DRA in 2014 [5], and the principle schematic is depicted in Figure 1. A detection probe labeled with a molecular marker is attached to a solid phase carrier (microplate, beads, resin, etc.) via streptavidin (SA)-biotin binding (Figure 1A). A single-stranded (ss) target DNA (i.e., dsDNA denatured to ssDNA, or cDNA) is added to the reaction solution and hybridizes to the probe forming dsDNA helix (Figure 1B). The probe–target hybrid carries a specific restriction site, thus the corresponding specific REase added to the reaction solution cleaves the helix. (Figure 1C). Upon cleavage, a part of the probe labeled with the molecular marker is released from the solid carrier into the reaction solution (Figure 1C). The solution is then transferred to a separate detection compartment and quantified (Figure 1D). Previously [5], we used horseradish peroxidase (HRP) as the molecular marker and quantified the signal optically by TMB (3,3',5,5'-Tetramethylbenzidine) oxidation at 655 nm. However, a variety of molecular markers can be used for this assay together with a wide range of detection techniques including fluorescent and electrochemical ones.

The developed DRA demonstrated the limit of detection of 1 nM with the dynamic range up to 30 nM [5]. The first assay was used for detection of methicillin-resistant *Staphylococcus aureus*, a bacterium with antibiotic resistance (MRSA). The assay was designed to detect a fragment of the *mecA* gene that has very high conservation (nearly 100% identity over 2 kb length) among various MRSA strains. A 40-mer probe MCA-BG (CAATTAAGTTTGCATAAGATCTATAAATATCTTCTTTATG) was designed from the *mecA* sequence commonly used for qPCR [6]. The central part of the probe had the specific recognition sequence (AGATCT) for BglII REase.

The assay was used to analyze (i) REase requirements for minimum target–probe helix sufficient for cleavage and signal generation, and (ii) the enzyme tolerance of mismatches and insertions. Our data showed a significant decrease in the assay signal when the probe–target length was reduced to 20-mer, with drastic reduction to nearly zero at the length of 16-mer. This length requirement suggested very high specificity, since on average in a random DNA sequence, a cognate 16-mer would be observed only once every 4.3 Gbp. We further analyzed the effects of mutations and showed that even a single mismatch within the restriction site eliminated the assay signal completely. In contrast, small (up to 3) target–probe mismatches and insertions (ssDNA loops) outside of the restriction site in the flanking sequences did not produce strong effects [5].

We concluded that the REase enzymatic cleavage in the process of DRA requires: (i) perfect probe–target match within a restriction site and (ii) at least 16-mer (preferably >20) of a hybridized dsDNA target–probe sequence around the restriction site. This study has been performed using BglII REase [5], with the caveat that other enzymes may be different in terms of mismatch and insertion tolerance.

The developed DRA method requires the ssDNA targets. In our previous work with dsDNA amplicons [5], heat denaturation of 95 °C was applied, followed with incubation on ice and addition to SA-coated microplate wells carrying pre-attached biotinylated probes. Alternatively (Figure 2), the same heat denaturation can be applied to a mixture of probe and target DNA in solution. The probe used in this approach contains the biotinylated target-specific part, and an oligonucleotide tag. The tag is used for subsequent attachment of the molecular marker HRP. HRP is covalently linked to an oligonucleotide complementary to the tag and is attached through DNA–DNA hybridization (Figure 2). Thus, after the probe–target reaction solution has been denatured and cooled down, it is mixed with the tagged HRP and added to the SA-coated solid carrier (Figure 2). This leads to the quick binding of the biotinylated probe to surface SA that occurs simultaneously with the probe–target and probe–HRP tag hybridization (Figure 2). After washing to remove unbound molecules, a specific corresponding REase is added to perform enzymatic cleavage (Figure 2). The resultant cleaved HRP released into the reaction solution is then quantified colorimetrically.

Since no signal amplification is employed for the DRA platform, it provides for the nanomolar range sensitivity. The resultant practical applications are limited to analysis of amplicons for a simple and inexpensive version of semi-quantitative PCR, and to detection of precultured microbial pathogens. The former is described in [5], and the latter is currently being developed in cooperation with Paradigm Diagnostics, Inc. (http://pdx-inc.com). Paradigm Diagnostics has a technology for the detection of numerous pathogens based on culturing food industry samples in media that change color in the presence of growing microorganisms. This approach permits to detect samples with live microbes; however, the pathogen presence needs to be confirmed by an independent molecular method.

We used DRA to develop a technique to detect pre-cultured Shiga toxin-producing *E.coli* strains. Typically, USDA recommends qPCR testing of these strains using two genes, Eae and Stx, with a well-characterized set of corresponding primers and probes [7]. We used the qPCR probe sequences to develop DRA probes, namely Stx: CTGGATGATCTCAGTGGGCGTTCTTATGTAA and Eae: ATAGTCTCGCCAGTATTCGCCACCAATACC. The probes contain the restriction sites CTCAG and CCAGT for specific cleavage with BspCNI and BsrI REases, respectively.

The developed assay technique is based on the scheme shown in Figure 2. The SA-coated microplates were used as a solid carrier for the probes. Inoculated food samples were precultured for 5–6 h and used for total DNA extraction. The resultant DNA samples were directly used for DRA without PCR amplification. Thus, the full assay time was below 1 h including probe–target hybridization and binding to the plate (20 min) and REase cleavage (20 min). Figure 3A,B shows the results of the Eae and Stx gene detection in sample sets precultured for 5 and 6 h, respectively. In both cases, the signal obtained for inoculated samples was significantly higher than that for a negative control. Thus, the DRA technique can provide a simple, low-cost and fast alternative to PCR-based molecular detection of foodborne pathogens in precultured samples that can be carried out with minimum equipment requirements in field laboratories.

## 3. Restriction Cascade Exponential Amplification (RCEA)

Restriction cascade exponential amplification (RCEA) has been introduced in 2015 [8]. A principle schematic of the assay is shown in Figure 4. It starts with the initial recognition stage that involves a target-specific probe modified with biotin at one end and an “amplification REase” molecule at the other end. The probe is attached to a solid carrier via SA–biotin interaction (Figure 4A). When the probe hybridizes with the corresponding target, the added free “recognition REase” cleaves the target–probe hybrid, releasing the amplification REase from the surface into the reaction solution and thus completing the first recognition stage (Figure 4B,C).

The reaction solution containing the released amplification REase is then transferred to the next amplification stage (Figure 4D). The corresponding setup contains amplification probes immobilized on a solid surface through biotin–SA interaction. The solution end of each probe is attached to the same amplification REase, as employed at the initial stage. In addition, an HRP molecule is attached to the solution probe end through complementary oligonucleotide tag hybridization (Figure 4D). The dsDNA amplification probes carry the specific restriction sites for cleavage with the attached amplification REases. However, the surface immobilization and double helix structure limit the attached REases’ mobility, making them incapable to bend and cut at the restriction site.

Addition of the reaction solution from the recognition stage that contains **free** molecules of amplification REase results in cleavage of the immobilized amplification probes and release of an additional molecule of amplification REases into the reaction solution (Figure 4D,E). Thus, each cleavage event doubles the amount of free amplification REases, resulting in a cascade of cleavage reactions. In addition, each cleavage event releases immobilized HRP markers into the reaction solution (Figure 4F). The released HRP can be measured, i.e., by transferring the reaction solution to a detection cell. The described amplification setup can be common for all RCEA assays, with the target specificity determined during the initial recognition step by using the specific recognition probe and recognition REase.

Our published study [8] demonstrated highly sensitive detection of the target mecA gene related to MRSA infections. We used the same combination of recognition probe and REase: 40-mer MCA-BG and BglII, as for DRA [5]. The amplification stage was designed using two amplification REases: BamHI (restriction site GGATCC) and EcoRI (restriction site GAATTC). The most serious challenge in the RCEA assay development was associated with conjugation of REase molecules with oligonucleotide probes. All commercially available enzymes lost their enzymatic activity during standard conjugation via amino groups. Similar results were reported in the literature [9]. Successful conjugation could only be achieved by using mutant enzymes (BamHI and EcoRI) that had been engineered for ligand attachment by replacing some surface “non-essential” amino acid residues with cysteines [9].

The MRSA RCEA assay was tested using a specific target oligonucleotide complementary to the MCA-BG. As shown in Figure 5, both amplification REases, BamHI and EcoRI, demonstrated similar performance with the lower detection limit of 10 aM concentration, and the linear dynamic range (at the logarithmic scale) up to 1 nM. The plot obtained for the same target oligonucleotide without amplification using DRA is shown at the right side of the Figure 5. The data show that the RCEA assay format gained the detection limit improvement of approximately eight orders of magnitude over the DRA. The data were obtained for non-optimized assay conditions, and we could still detect as little as 200 target molecules per sample. This performance is similar to the detection limit of PCR applications and can likely be further improved by RCEA assay optimization. However, the main goal of such optimization should be the overall assay time that currently stays at about 2 h and can be significantly reduced to less than 1 h by improvement of mass transfer in the two-phase (liquid and solid) system. The improvement of mass transfer can be achieved by agitation and mixing, optimization of cell geometry, increase of surface to volume ratio, etc.

## 4. Tandem Oligonucleotide Repeat Cascade Amplification (TORCA)

An attractive alternative format for REase-based signal amplification employs another type of a trigger that is an unmodified oligonucleotide rather than an REase enzyme molecule. The obvious advantage is the omission of the REase conjugation step, enabling the use of standard commercially available enzymes at all stages of the assay. The first developed assay used two species of amplification trigger oligonucleotides, Tr1 and Tr2, that can start a self-perpetuating cascade of REase-catalyzed events based on trigger hybridization with each other single-stranded linker.

The assay based on this principle, tandem oligonucleotide repeat cascade amplification (TORCA), was introduced in 2019 [10]. This format employs standard REases that are suspended in the reaction solutions without immobilization. To prevent cleavage events, the restriction sites of amplification probes are kept single-stranded, and the reaction cascade is started by addition of a free trigger oligonucleotide released during the initial recognition reaction. The exponential amplification is then achieved by usage of several tandem repeats of the same trigger oligonucleotide within each probe.

The principle schematic of the TORCA assay is shown in Figure 6. It starts with the recognition step involving an oligonucleotide recognition probe specific for a target of interest that is extended with the “trigger” oligonucleotide unit Tr1 (Figure 6A). The probe is immobilized on a solid surface through biotin–SA interaction. Upon the target–probe hybridization (Figure 6B), the resultant dsDNA is cleaved with the corresponding recognition REase that is present in the reaction solution (Figure 6C). This cleavage releases the trigger Tr1 into the reaction solution at the amount proportional (ideally, equal) to the amount of the target added.

At the next amplification stage, the recognition reaction solution is transferred to an amplification chamber that contains two types of amplification probes immobilized on a solid carrier (Figure 6D). Each probe has HRP attached to the solution end. The amplification probe AP1 consists of a sequence complementary to the trigger Tr1 (aTr1) attached to the carrier surface and **multiple** tandem repeat sequences of trigger Tr2 at the solution end. The amplification probe AP2 has a sequence complementary to Tr2 (aTr2) at the surface and multiple Tr1 sequences at the solution end (Figure 6D). The amplification chamber also contains two free amplification REases that specifically cleave the dsDNA hybrids of Tr1-aTr1 and Tr2-aTr2. Since initially all probes are present in the chamber as single-stranded, no enzymatic cleavage is observed. Addition of the recognition reaction solution containing free Tr1 results in hybridization with the complementary aTr1 part of AP1, followed by the cleavage and release of multiple Tr2 into the solution (Figure 6 D,E). In turn, the released Tr2 molecules hybridize to the immobilized complementary aTr2 within AP2, resulting in the further cleavage and release of numerous Tr1 units. Since each cleavage event is accompanied with the release of multiple trigger units and thus initiates the cleavage of the next amplification probes (Figure 6F), this process is self-perpetuating and provides for exponential accumulation of unbound HRP and thus the exponential assay signal increase over time.

Unlike RCEA, the TORCA assay format does not require REase conjugation, regular commercially available enzymes can be used at all stages. Moreover, the recognition probes do not contain the attached enzyme, thus they can be safely subjected to denaturation at high temperature. This is an important advantage to streamlining the initial recognition step: instead of separate denaturation of dsDNA targets before mixing and hybridization with recognition probes, the targets and probes can be mixed, denatured and hybridized in a single step (similar to the DRA scheme shown in Figure 2).

The main challenge for TORCA assay development is associated with prevention of physical contacts between the amplification probes AP1 and AP2. Any contact will lead to hybridization of the complementary Tr and aTr units followed by cleavage, and thus initiation of the amplification cascade without addition of a free trigger. Indeed, two types of beads, modified with either AP1 or AP2, when mixed in the presence of both amplification REases, immediately start releasing some HRP signal [10]. At the same time, if only one amplification REase is present, the HRP release does not occur. One possible solution is membrane separation, and our data showed that such physical separation of the beads with AP1 and AP2 prevents the HRP release in the presence of both amplification REases [10].

Based on these data, we designed two types of amplification chambers for the TORCA assays. The first type employs a mixture of two probe carriers without physical separation. In this chamber, addition of a trigger from the recognition step enhances the rate of HRP signal generation over a rather prominent background of the trigger-independent HRP release. The obvious disadvantage is high background values that need to be carefully measured with negative controls. The main advantage of this approach is the short assay time, approximately 15 min for the whole amplification stage [10]. The second type of amplification chamber provides for the physical separation of two different probe carriers with a membrane permeable for DNA molecules but not for carrier particles. This type is associated with a low background; however, it requires a considerably longer time for the completion of the amplification stage (over 1 h).

The two approaches have been tested in [10] using SspI (restriction site AATATT) and EcoRV (restriction site GATATC) as amplification REases. The amplification probes had seven repeats of Tr1 and Tr2. Figure 7 shows the TORCA data obtained for trigger detection using the two amplification formats: mixture of non-separated probes (Curve b) and membrane-separated probes (Curve c) [10]. They are compared to curve “a” obtained for the same trigger without amplification by using DRA.

Both TORCA formats had the same detection limit of 10 aM concentration similar to RCEA and to PCR applications. The format with the mixture of non-separated probes demonstrated a little less sensitivity as compared with the membrane-separated probes, however, it used a shorter amplification time and showed better linearity. Thus, the probe mixture assay format was chosen for further development of an assay to detect malaria *P*. *falciparum* parasites by using RNA as a target.

This RNA-based approach was a step toward the goal to distinguish between past and ongoing malaria infections. Such discrimination performed directly at point-of-care facilities is essential to direct drug therapy at only those patients who can benefit from it, and to conduct new drug clinical trials in malaria-endemic areas [10]. RNA stability is known to be significantly lower than stability of DNA and protein malaria markers, thus RNA detection is likely to better correlate with the presence of a live parasite as compared with stable DNA. Since REase enzymes can only cleave DNA–DNA hybrids, RNA was reverse-transcribed into cDNA.

The developed TORCA assay format was compared to PCR detection [10]. The TORCA assay sensitivity toward three different malaria RNA targets had the detection limit of about 7.5 IE per 100 μL of blood sample. This is almost two orders of magnitude better than the malaria detection limit recommended by WHO [11]. The observed linear dynamic range for the assay spanned approximately three orders of magnitude.

Direct comparison of the TORCA assay versus common RT PCR is presented in Figure 8. Both methods successfully detected the *P. falciparum* parasite RNA targets at different times after initiation of the drug treatment of a patient [10]. The decrease of parasite RNA directly correlated with the post-treatment time, and both methods showed considerable consistency (Figure 8). The described method of distinguishing between past and ongoing infections is based on observations showing much lower stability of pathogen RNA as compared with DNA. However, for each particular infection, an independent study is needed to establish a correlation between target RNA content and disease state.

An important feature of the TORCA assay is the ability to simultaneously detect multiple targets in the same chamber, generating an integrated signal to enhance assay sensitivity and to provide for mutation tolerance. For malaria detection, we successfully employed three different RNA targets, and thus three recognition probes in the same assay [10].

The TORCA optimization efforts are currently focused on finding a middle ground in the design of the amplification chamber: to combine the separation membrane to reduce backgrounds with the enhanced mass transfer to shorten the assay time. The latter can be done by using an increased membrane surface and physical agitation of the carrier particles.

An important advantage of the TORCA and RCEA formats is associated with their high tolerance of various target sizes. In contrast to PCR-based assays that normally require at least 70 bp fragments, the TORCA and RCEA minimum size requirement is 20 bp. Since the assay can be used for recognition of very short fragments, the potential applications include working with partially degraded nucleic acids in FFPE material and in liquid biopsies containing cell-free DNA and RNA from serum and plasma. Archived FFPE tissues are subjected to formalin-induced crosslinking of nucleic acids to proteins, base purination and strand breaks. As a result, the proportion of RNA fragments <200 bases is typically >50%, and can be as high as 90%, making these samples unsuitable for standard assays that require templates >150 bases. In contrast, REase-based assays do not have the 150-base size limitation [12].

In contrast to the RCEA technology, TORCA does not require enzyme engineering and complex conjugation. All required components, including custom oligonucleotides, are commercially available at low cost with a fast turn-around time, thus ensuring great flexibility towards the development of new assays towards emerging targets of interest.

## 5. REase Based Assays: Advantages and Limitations

All REase-based assays described above [5,8,10] have several common advantages associated with high specificity due to the “double-proof” combination of two biorecognition events. They are nearly insensitive to (i) non-specific and partially complementary DNA binding, (ii) excess foreign DNA background and (iii) various non-nucleic acid contaminants (such as proteins, PCR inhibitors, etc.).

The isothermal nature of REase-driven assays, their simplicity and flexibility provide opportunities for the development of assay cartridges with all components included. The assays can utilize various molecular markers of different nature and use colorimetric, fluorescent and electrochemical detection formats without the need of complex instrumentation, and at much lower costs as compared with other nucleic acid assays. Both fluorescent and electrochemical detection formats would allow for the direct detection of the released molecular label in the amplification chamber in real time, without the need to transfer the reaction solution to a special detection chamber. To get rid of this end-point fluid transfer, and to switch to a kinetic assay mode, one needs to place an electrode or optical detection probe into the amplification chamber and physically separate it from the solid carrier with immobilized assay reagents. This can be achieved by various engineering approaches, providing for design flexibility, and enabling the development of devices for field and point-of-care applications.

The REase-based assays provide for easy adaptation to new analytes of interest. This adaptation involves a single design step to develop a pair of a 20–30 bp recognition probes with a corresponding recognition REase. This simplicity provides for a great advantage over another isothermal assay, LAMP, that requires the use of 4–6 carefully optimized primers for each new DNA target. The current selection of commercially available REases with different restriction site sequences is expanding continuously. New England Biolabs alone offers over 285 restriction enzymes. In our experience, any sufficiently diverse 100–200 bp coding DNA sequence typically contains at least one option for a possible recognition probe with an REase restriction site. Further, in contrast to the design of recognition stage components, the same amplification stage reagents can be used for all targets of interest.

The two highly sensitive exponential amplification assay formats, RCEA and TORCA, achieve the attomolar detection limit similar to the golden standard of PCR. However, in contrast to PCR, they amplify the assay signal rather than the DNA template. The main issue of template DNA amplification is a possible sequence-dependent bias that is widely observed in PCR, when certain sequences have much higher amplification efficiency than the other [13]. The REase-based signal amplification does not employ the production of multiple new DNA copies, thus it is free of sequence-dependent bias and mis-priming issues.

The biggest challenge in the development of REase-based exponential signal amplification assays is the engineering of automated fluid transfer. Currently, reaction solutions are transferred manually, however, the operator involvement needs to be minimized for point-of-care and field applications. This will provide a competitive edge for the REase-based assays that can serve as a valid alternative to PCR due to their simplicity, low cost, high specificity and sensitivity of detection.

## 6. Conclusions

Our perspectives on applications of the REase-based nucleic acid assays are associated with their versatility, low cost, simplicity, specificity, isothermal nature and potential for the development of portable automated instrumentation formats. The two highly sensitive assay formats based on exponential amplification are valid alternatives to PCR-based assays. They can be performed at a fraction of the PCR cost in low-resource settings, including point-of-care laboratories, and field facilities. We believe that the described formats have high potential for biosensors development, since they can use both fluorometric and electrochemical detection. The detection of a released molecular label can be done in real time, directly in the amplification chamber during the amplification stage, with no extra fluid transfer steps, and employing a kinetic assay mode. This allows for substantial simplification of the assay setup and supporting instrumentation along with a reduction in the assay time. In our perspective, the REase-based platforms also overcome the very serious limitation of nucleic acid assays, namely the minimum target length requirement that is important for both PCR and isothermal formats such as LAMP. Thus, the novel platforms have potential for considerable expansion into new niches involving the analysis of highly fragmented nucleic acids, including liquid biopsies. However, further development of the described technologies requires optimization in terms of simplification, automation, robustness and short assay time.

## Figures and Tables

**Figure 1 sensors-20-03873-f001:**
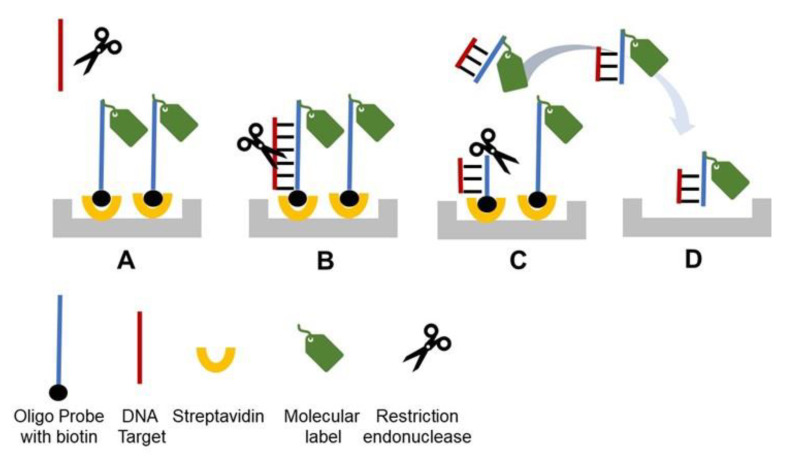
General schematic of the direct restriction assay (DRA). (**A**) A molecular marker/label is conjugated to an oligonucleotide probe that is specific for a target gene of interest and immobilized on a solid surface through biotin-SA binding. (**B**) Target DNA (an oligonucleotide or denatured dsDNA) is hybridized to the immobilized probe. (**C**) A restriction enzyme recognizes and cleaves the target–probe dsDNA hybrid, resulting in the release of the molecular marker into the reaction solution. (**D**) The reaction solution is transferred into a new well to quantify the molecular marker. For each target DNA molecule, one molecular marker is released, resulting in linear dependence between the assay signal and the target DNA concentration.

**Figure 2 sensors-20-03873-f002:**
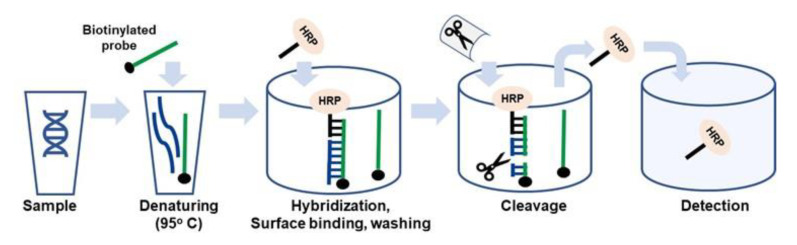
General schematic of the new approach to probe–target hybridization for DRA. A sample containing dsDNA targets is supplemented with a specific biotinylated probe and subjected to DNA denaturation at 95 °C followed by quick incubation on ice. The denatured probe and target mixture are supplemented with horseradish peroxidase (HRP) covalently attached to an oligonucleotide tag for hybridization to the probe. The mixture is added to the streptavidin (SA)-coated solid carrier for attachment and hybridization of the specific targets and tagged HRP to the probes. After washing to remove the unbound molecules, the specific REase is added, catalyzing enzymatic cleavage and HRP release. The free HRP is transferred to a detection cell.

**Figure 3 sensors-20-03873-f003:**
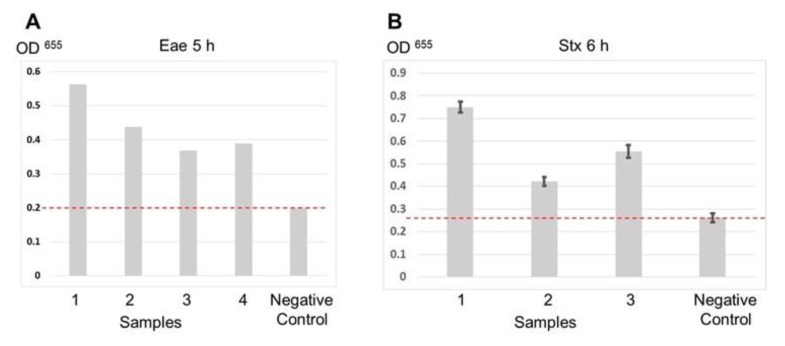
DRA data obtained sets of samples inoculated with Shiga toxin-producing *E.coli.* (**A**) Eae gene detection for samples precultured for 5 h. Data for Eae gene detection were obtained in singlicate. (**B**) Stx gene detection for samples precultured for 6 h. The dash lines indicate the signal level for negative control (non-inoculated samples).

**Figure 4 sensors-20-03873-f004:**
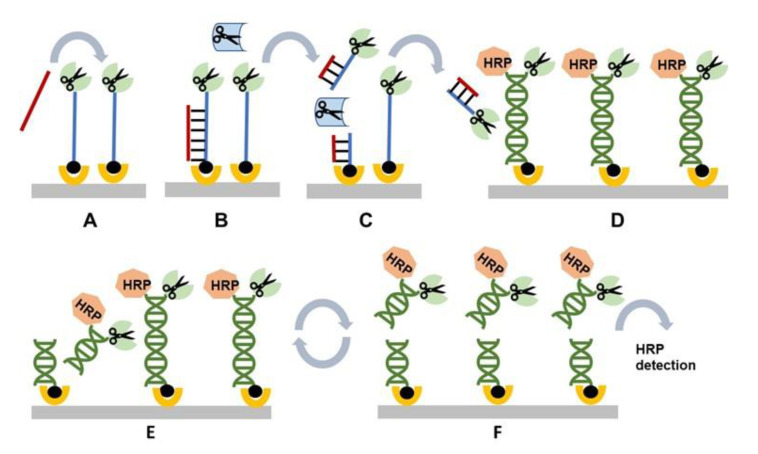
General schematic of the restriction cascade exponential amplification (RCEA) assay. (**A**) An oligonucleotide probe specific for a target of interest is conjugated to an REase for amplification and attached to a solid substrate using biotin. A test sample containing the target of interest is added. (**B**) The target in the test sample hybridizes to the probe. (**C**) The hybrid is specifically cleaved by a recognition REase. amplification REase is subsequently released into the solution. (**D**) The reaction solution is transferred to an amplification cell that contains an excess of immobilized amplification REase attached to the surface through an oligonucleotide linker. The linker contains the restriction site corresponding to the amplification REase, and it is double-stranded, with the second strand conjugated with HRP. All amplification REase molecules in the amplification cell are immobilized and thus incapable of cleaving their own or neighboring linkers. Addition of the free amplification REase generated in (**C**) triggers linker cleavage, releasing additional amplification REase, which in turn cleaves new linkers. (**E**) Each step of this exponential cascade of cleavage reactions doubles the amount of free amplification REase molecules in the reaction solution. (**F**) The linker cleavage releases HRP, which is quantified colorimetrically. Each initial target–probe hybridization event produces an exponentially amplified number of HRP molecules, with the value dependent on the amplification time.

**Figure 5 sensors-20-03873-f005:**
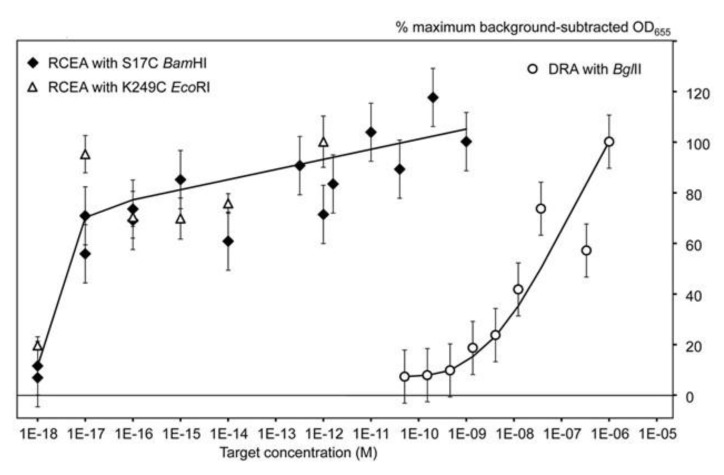
From [8] (Creative Commons CC-BY-NC-ND license). The RCEA limit of detection evaluated using the oligonucleotide target AMC-BG. The X-axis shows the target concentrations (M) and the Y-axis shows the background-subtracted HRP signal values (with the background calculated as the mean signal generated for zero target concentrations). For normalization and comparison of sample series, the HRP signal values were expressed as the percentages of the maximum background-subtracted OD_655_, corresponding to each series. Open circles show the data generated using the direct restriction assay (DRA) with no amplification. The other two series were generated using the RCEA assays with the mutant S17C BamHI (closed diamonds) and K249C EcoRI (open triangles) as amplification REases. Error bars show standard deviations.

**Figure 6 sensors-20-03873-f006:**
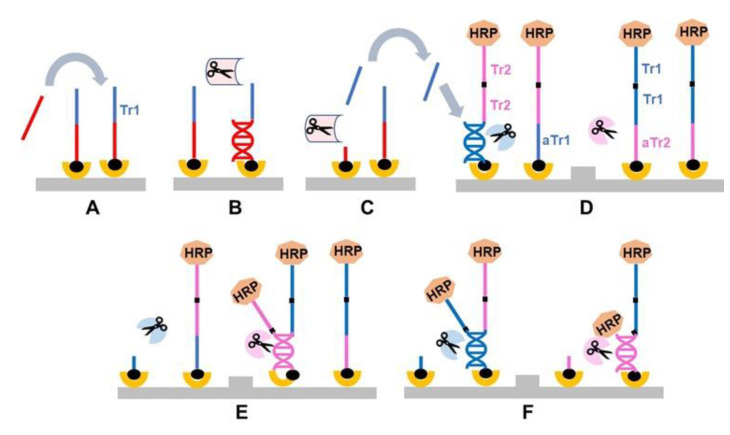
General schematic of tandem oligonucleotide repeat cascade amplification (TORCA). (**A**–**C**) The recognition stage: An oligonucleotide probe specific for a target of interest is extended with a “trigger” unit (Tr1) and attached to surface using biotin. A test sample containing the target of interest is added (A). The target in the test sample hybridizes to the probe (**B**), and the hybrid is specifically cleaved by a specific recognition REase (**C**). The Tr1 unit is subsequently released into the reaction solution. (**D**–**F**) The amplification stage: The reaction cell carries two types of amplification probes. The first contains a single unit complementary to the trigger sequence Tr1 (antisense Tr1, aTr1), and multiple identical units of a trigger sequence Tr2. The second contains multiple identical Tr1 units, and a single unit complementary to the Tr2 unit (antisense Tr2, aTr2). Both probe types are surface-attached and contain a molecular marker HRP on their solution-facing end (**D**). The reaction solution in the amplification chamber contains two common REases, specific to Tr1 and Tr2, that recognize and cleave dsDNA hybrids of Tr1-aTr1 and Tr2-aTr2, respectively. When the recognition reaction solution is transferred to the amplification cell, the free trigger Tr1 hybridizes to an aTr1 unit of the first probe leading to the probe cleavage by Tr1-REase (**D**) and release of Tr2 into the reaction solution (**E**). In turn, the released Tr2 hybridize to an aTr2 of the second probe type (**E**), causing cleavage of Tr2 and further release of additional Tr1 units. This cascade of events also results in the release of the HRP molecular marker that can be used for signal quantification (**F**).

**Figure 7 sensors-20-03873-f007:**
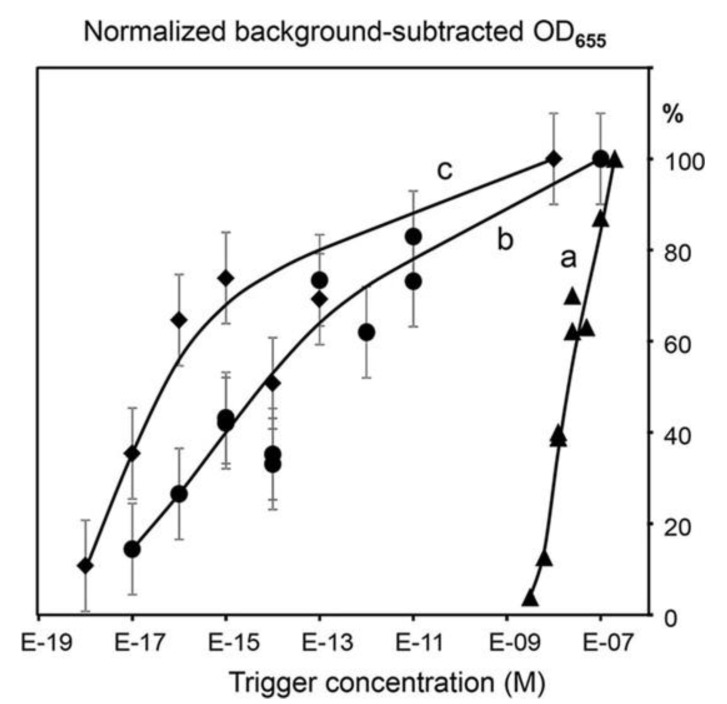
From [10] (Creative Commons CC-BY license). The dependence of the HRP-generated signal on the concentration of the amplification trigger added to the single (non-amplified DRA format) REase (**a**) or the two REases EcoRV and SspI (**b-c**) systems. The curves **a** and **b** are generated for mixtures of the two bead types, one modified with the amplification probe AP1-HRP and the other with AP2-HRP. The curve **c** was obtained for the same two bead types separated by a filter barrier. The X-axis shows the target concentrations (M), and the Y-axis shows the background-subtracted and normalized HRP signal values. The background was calculated as the mean signal generated for the triplicate no-trigger added negative controls. For normalization and comparison of the sample series, the HRP signal values are expressed as the percentages of the maximum background-subtracted OD_655_ corresponding to each series. Error bars show standard deviations. The data for (non-amplified DRA format) REase (a) were obtained without replicates.

**Figure 8 sensors-20-03873-f008:**
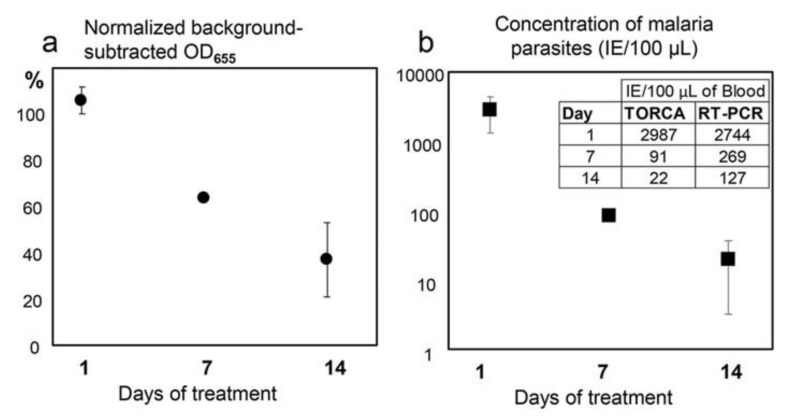
From [10] (Creative Commons CC-BY license). The dependence of the TORCA signal (**a**) and the calculated infected erythrocyte concentration (**b**) on the time after initiation of the patient drug treatment (X-axis, days). (**a**) The Y-axis shows the background-subtracted and normalized HRP signal values. The background was calculated as the mean signal generated for the triplicate no-trigger added negative controls. For normalization and comparison of the sample series, the HRP signal values are expressed as the percentages of the background-subtracted signal obtained for the positive control containing an equimolar mixture of targets at 10 nM concentrations. (**b**) The Y-axis shows the IE (Infected Erythrocytes) concentrations calculated using the standard calibration curve obtained separately. The inset shows mean data for IE concentrations measured using TORCA and reverse transcription-PCR methods and calculated according to corresponding calibration curves. Error bars in both graphs show standard deviations. Error bars for 7-day treatment are smaller than the marker.

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
