# Peer review of "Restriction Endonuclease-Based Assays for DNA Detection and Isothermal Exponential Signal Amplification"

_sensors, 2020, doi:10.3390/s20143873_

Round 1
Reviewer 1 Report
The Perspective on restriction endonuclease-based assays from Smith et al is a good overview of different approaches and methods for DNA detection with restriction enzymes. The authors clearly present the various versions of these assays and give a good comparison between them, honestly describing the limitations and advantages for a useful guide to the field. I have a few comments on things I think should be addressed more thoroughly, given the "Perspective" nature of the manuscript to giver a fuller picture.
1) The authors describe PCR and isothermal amplification methods as having drawbacks or limitations. But those methods are clearly not too limited, with widespread use in diagnostics, and isothermal methods in particular can use very simple instruments and workflows. So, I'm curious as to the advantages using restriction enzymes. Reaction time, requirement for liquid transfer, etc. don't seem to make these methods particularly advantageous. I buy specificity, but nucleic acid amplification approaches are pretty sensitive too. Sensitivity of some of the methods here is good, but PCR can get to <10 copies without too much trouble. The one big advantage seems to me to be the ability to detect very short targets, enabling detection where LAMP or PCR would fail. Highlighting more the use of FFPE samples or degraded eDNA/RNA would help point out why these methods are unique and useful.
2) It seems obvious to me that use of a Cas enzyme could enable direct RNA detection without RT, does require a PAM but otherwise any sequence could be targeted w/o need for an RE site. Would that work, could it be useful? I imagine it has been considered.
3) Commentary on what makes a useful RE would be good to add. There are >200 available commercially, but only a handful make it in here. Conjugation requirement is an obvious potential issue, but nature of cleavage (IIS vs. IIP) or temperature optimum would be good to discuss.
Author Response
Thank you very much for very useful comments. Please find below our answers to your critique. All changes made in manuscript are highlighted by a “Track Changes" function of Microsoft Word.
Point 1.
The authors describe PCR and isothermal amplification methods as having drawbacks or limitations. But those methods are clearly not too limited, with widespread use in diagnostics, and isothermal methods in particular can use very simple instruments and workflows. So, I'm curious as to the advantages using restriction enzymes. Reaction time, requirement for liquid transfer, etc. don't seem to make these methods particularly advantageous. I buy specificity, but nucleic acid amplification approaches are pretty sensitive too. Sensitivity of some of the methods here is good, but PCR can get to <10 copies without too much trouble. The one big advantage seems to me to be the ability to detect very short targets, enabling detection where LAMP or PCR would fail. Highlighting more the use of FFPE samples or degraded eDNA/RNA would help point out why these methods are unique and useful.
Response to point 1.
We agree with the reviewer that at this point PCR and LAMP based methods are widespread in diagnostics. However, we do not position the restriction endonuclease-based formats as substitution for the conventional nucleic acid assay methods. We believe that our technologies will complement the existing methods by finding certain niches, where their combination of advantages (low cost, short target size, ability to perform at non-laboratory conditions) will have high significance.
We agree that the liquid transfer stages represent a challenge for practical applications of our methods. However, there are other methods, such as ELISA, that employ multiple fluid manipulation steps and still have a wide range of practical applications. We understand the need for a fully automated assay format with alternative detection methods (electrochemical or fluorescent), and mention this point in sections 5 and 6 of the manuscript (lines 439-443 and 466-467).
We agree the need to highlight the use of FFPE samples, and added extra information to the manuscript: Archived FFPE tissues are subjected to formalin-induced crosslinking of nucleic acids to proteins, base purination and strand breaks. As the result, the proportion of RNA fragments < 200 bases is typically >50%, and can be as high as 90%, making these samples unsuitable for standard assays that require templates >150 bases. In contrast, REase-based assays do not have the 150 base size limitation [12]. (section 4 lines 396-400)
- Patel, P.G.; Selvarajah, S.; Guérard, K-P.; Bartlett, J.M.S.; Lapointe, J.; Berman, D.M.; et al. (2017) Reliability and performance of commercial RNA and DNA extraction kits for FFPE tissue cores. PLoS ONE, 2017, 12(6): e0179732. https://doi.org/10.1371/journal.pone.0179732 (references, lines 504-506).
Point 2.
It seems obvious to me that use of a Cas enzyme could enable direct RNA detection without RT, does require a PAM but otherwise any sequence could be targeted w/o need for an RE site. Would that work, could it be useful? I imagine it has been considered.
Response to point 2.
Application of Cas enzymes has been described in the literature and considered by us as a version of the assays described in the manuscript. However, at this point we do not have experimental data on the use of this system, and thus, would prefer not to add this discussion into the manuscript.
Point 3.
Commentary on what makes a useful RE would be good to add. There are >200 available commercially, but only a handful make it in here. Conjugation requirement is an obvious potential issue, but nature of cleavage (IIS vs. IIP) or temperature optimum would be good to discuss.
Response to point 3.
New England Biolabs alone offers over 285 restriction enzymes (https://www.neb.com/products/restriction-endonucleases/restriction-endonucleases). We have added this information to the manuscript to enhance good commercial availability of REases (section 5 line 428). Our experimental work employed only the ‘orthodox’ Type IIP REases that recognize and cleave a single palindromic DNA recognition sequence 4 to 8 bp-long (Mucke M, Kruger DH, Reuter M. Diversity of type II restriction endonucleases that require two DNA recognition sites. Nucleic Acids Res. 2003;31(21):6079-6084. doi:10.1093/nar/gkg836). Their performance does not depend on the length of the recognition site. However, the use of enzymes with long 6-8 bp recognition sequences makes DNA target fragments less likely to contain extra cleavage sites, and thus, help to simplify the probe design. On the other hand, our data showed that even short 4-bp restriction site does not reduce the assay specificity.
We are planning to test the Type IIS REases in near future, and since we do not yet have experimental data, we would prefer not to discuss them in the manuscript.
Conjugation issues present a big challenge and are discussed in the manuscript (section 3 lines 229-235) and is discussed in detail in the original publication. However, it is only needed for an amplification enzyme, and this REase can be common for all RCEA assays, since the assay specificity is determined during the initial recognition step using non-conjugated enzyme. The main limitation of using a single amplification REase in all assays is that a recognition probe should not contain the corresponding restriction site. This limitation can be circumvented by having not one but two common amplification enzymes, either of which can be employed at the amplification stage depending on target requirements.
Temperature optimum is clearly an important assay parameter. Thus far, we intentionally selected REases that function at 37 C for assay simplicity. However, employment of enzymes with elevated optimum temperatures can be a good way for to improve the assay performance for GC-rich targets in the future.
Reviewer 2 Report
The presented perspective on REase-based nucleic acid detection is well written and will be useful to numerous new users of this application.
I have only few minor points listed below:
- Figure 3: Error bars should be included (A) or state in the legend that the readings are from single samples. Indicate whether the differences in samples compared to negative control are statistically significant.
- Figure 5: Curves needed for both BamHI and EcoRI.
- Figure 7: Error bars on curve a or indicate if no replicates.
- Line 350 -355: explain bit more about distinguishing past and present infections. Can these germs be dormant, i.e. present without RNA expression...etc.
- Figure 8: why no error bars on 7-Day sample(s)? Also indicate statistical significance.
- Line 382 -385: can centrifugation be used to increase mixing?
- Line 391 - 393: comment on microRNA detection by REase system in comparison with other methods.
- Finally, it will be also useful to compare REase system with the new ones based on CRISPR. Cas9 variants can also be used instead of REase and consider flexibility, cost...etc.
Author Response
Thank you very much for very useful comments. Please find below our answers to your critique. All changes made in manuscript are highlighted by a “Track Changes" function of Microsoft Word.
Point 1.
Figure 3: Error bars should be included (A) or state in the legend that the readings are from single samples. Indicate whether the differences in samples compared to negative control are statistically significant.
Response to point 1.
The data in Figure 3 (A) was obtained without replicates. The following sentence was added to the Figure legend: “Data for Eae gene detection was obtained in singlicate” (section 2 line 171).
Point 2.
Figure 5: Curves needed for both BamHI and EcoRI.
Response to point 2.
Figure 5 is a reproduction of the original publication in Scientific Reports. We think that making changes to this Figure will make it more difficult for the readers to go back and get the extra details from the original paper.
Point 3.
Figure 7: Error bars on curve a or indicate if no replicates.
Response to point 3.
The data in Figure 7 plot a was obtained without replicates. The following sentence was added to the Figure legend: “The data for (non-amplified DRA format) REase (a) was obtained without replicates.” (section 4 lines 345-346).
Point 4.
Line 350 -355: explain bit more about distinguishing past and present infections. Can these germs be dormant, i.e. present without RNA expression...etc.
Response to point 4.
The described method of distinguishing between past and ongoing infections is not a universal approach. Although RNA stability is known to be significantly lower than that of DNA, an independent study is needed for each particular infection to establish correlation between target RNA content and disease state. To clarify this point we added additional discussion to the manuscript: “The described method of distinguishing between past and ongoing infections is based on observations showing much lower stability of pathogen RNA as compared to DNA. However, for each particular infection an independent study is needed to establish correlation between target RNA content and disease state.” (section 4, lines 367-370)
Point 5.
Figure 8: why no error bars on 7-Day sample(s)? Also indicate statistical significance.
Response to point 5.
The sentence “Error bars for 7-day treatment are smaller than the marker” is added to the Figure 8 caption (section 4, line 382)
Point 6.
Line 382 -385: can centrifugation be used to increase mixing?
Response to point 6.
We avoid centrifugation stages to reduce the assay complexity and to eliminate the need for extra instruments. To improve mass transfer we currently use rotation, and further on we are planning to use controlled electromagnetic mixing of magnetic beads.
Point 7.
Line 391 - 393: comment on microRNA detection by REase system in comparison with other methods.
Response to point 7.
We thank the reviewer for pointing out that our statement on microRNA detection is not supported experimentally. This statement has been deleted from the manuscript.
Point 8.
Finally, it will be also useful to compare REase system with the new ones based on CRISPR. Cas9 variants can also be used instead of REase and consider flexibility, cost...etc.
Response to point 8.
We agree that it will be useful to compare REase system with the detection systems based on CRISPR. However, both systems are still at the early stage of development, thus, we intentionally avoided such comparison until more details emerge regarding the proposed CRISPR/Cas formats. Moreover, we tried not to focus on comparison of REase based assays with numerous conventional nucleic acid assays, but rather to point out potential advantages and limitations of our formats. The conventional assay methods are mostly mentioned in the manuscript to have a golden standard and a reference point.
Reviewer 3 Report
The manuscript described a combination of isothermal exponential signal amplification with specific target DNA fragmentation assay. The presented results showed some advantages of the combination as described in other references. It would be interested by some readers who is developing cascade amplification techniques.
More detail about the condition and comparison with other related methods should be described in a table or text.
Author Response
Thank you very much for very useful comments. Please find below our answers to your critique. All changes made in manuscript are highlighted by a “Track Changes" function of Microsoft Word.
Point 1.
More detail about the condition and comparison with other related methods should be described in a table or text.
Response to point 1.
We agree that more details on assay conditions and comparison with other related methods would be useful. However, this manuscript is aimed to provide a general summary and overview of our novel formats that have been mostly published elsewhere. The original publications (that are all published in Open Access journals) contain detailed descriptions of protocols, experiments and data. Therefore, readers who are interested in extra information could easily obtain it. In contrast, this manuscript can serve as an introduction of our new technologies.